# Infant Gut Microbiota Associated with Fine Motor Skills

**DOI:** 10.3390/nu13051673

**Published:** 2021-05-14

**Authors:** Inmaculada Acuña, Tomás Cerdó, Alicia Ruiz, Francisco J. Torres-Espínola, Ana López-Moreno, Margarita Aguilera, Antonio Suárez, Cristina Campoy

**Affiliations:** 1Department of Biochemistry and Molecular Biology, Faculty of Pharmacy, Campus of Cartuja, University of Granada, 18071 Granada, Spain; iacuna@ugr.es; 2Center of Biomedical Research, Institute of Nutrition and Food Technology “José Mataix”, University of Granada, Armilla, 18016 Granada, Spain; alopezm@ugr.es (A.L.-M.); maguiler@ugr.es (M.A.); 3Carlos III Health Institute, Avda. Monforte de Lemos 5, 28029 Madrid, Spain; tomas.craez@isciii.es; 4Centre for Inflammation Research, Queen’s Medical Institute, University of Edinburgh, Edinburgh EH16 4TJ, UK; aruizro@ed.ac.uk; 5EURISTIKOS Excellence Centre for Paediatric Research, Biomedical Research Centre, University of Granada, 18071 Granada, Spain; fjtespinola@yahoo.es (F.J.T.-E.); ccampoy@ugr.es (C.C.); 6Department of Microbiology, Faculty of Pharmacy, Campus of Cartuja, University of Granada, 18071 Granada, Spain; 7Department of Paediatrics, School of Medicine, University of Granada, 18071 Granada, Spain; 8Spanish Network of Biomedical Research in Epidemiology and Public Health (CIBERESP), Granada’s Node, Institute of Health Carlos III, 28029 Madrid, Spain

**Keywords:** microbiota, neurodevelopment, gut–brain axis, fine motricity, probiotics

## Abstract

BACKGROUND: During early life, dynamic gut colonization and brain development co-occur with potential cross-talk mechanisms affecting behaviour. METHODS: We used 16S rRNA gene sequencing to examine the associations between gut microbiota and neurodevelopmental outcomes assessed by the Bayley Scales of Infant Development III in 71 full-term healthy infants at 18 months of age. We hypothesized that children would differ in gut microbial diversity, enterotypes obtained by Dirichlet multinomial mixture analysis and specific taxa based on their behavioural characteristics. RESULTS: In children dichotomized by behavioural trait performance in above- and below-median groups, weighted Unifrac b-diversity exhibited significant differences in fine motor (FM) activity. Dirichlet multinomial mixture modelling identified two enterotypes strongly associated with FM outcomes. When controlling for maternal pre-gestational BMI and breastfeeding for up to 3 months, the examination of signature taxa in FM groups showed that *Turicibacter* and *Parabacteroides* were highly abundant in the below-median FM group, while *Collinsella*, *Coprococcus*, *Enterococcus*, *Fusobacterium*, *Holdemanella*, *Propionibacterium*, *Roseburia*, *Veillonella*, an unassigned genus within *Veillonellaceae* and, interestingly, probiotic *Bifidobacterium* and *Lactobacillus* were more abundant in the above-median FM group. CONCLUSIONS: Our results suggest an association between enterotypes and specific genera with FM activity and may represent an opportunity for probiotic interventions relevant to treatment for motor disorders.

## 1. Introduction

In an infant, two concurrent biological processes take place during the first years of life—the foundational period of gut microbial colonization and the formation and refinement of neural networks responsible for a vast repertoire of behaviours characteristic of early life [1,2,3]. The assembly of the gut microbiota starts from birth, characterized by a rapid rate of colonization and expansion of gut bacteria dominated by *Actinobacteria* and *Proteobacteria* shifting towards a community dominated by *Firmicutes* and *Bacteroidetes*, a maturation process that coincides in time with the intense synaptogenesis and pruning in the cerebral cortex, ending in adolescence [4,5]. Experiments in animal models have shown that these two processes are physiologically connected. Experimental manipulations that alter the intestinal microbiota impact anxiety- and depression-related behaviours [5,6,7], and cognitive effects have also been reported [8,9]. Early gut establishment appears to be of critical importance because bacterial colonization of germ-free mice does not normalize anxiety phenotypes at 10 weeks of age [10], but does normalize anxiety phenotypes when the microbiota is introduced earlier in life, at 3 weeks of age [11]. Behavioural changes are accompanied by changes in neurochemistry, gene expression, dendritic morphology, and subcortical brain volumes [7,11,12,13,14,15,16]. Germ-free mice also display an increased rate of turnover of noradrenaline, dopamine, and 5-HT in the brain [7], possibly responsible for their well-documented increased motor activity, since these neurotransmitters have roles in increasing blood flow to muscle and central motor control, respectively.

In infants, this mechanistic relationship has not been empirically demonstrated but several associations between gut microbiota and behaviour have been reported. Altered microbial composition of the gut has been reported in children with autism [17,18,19], and it is also linked to childhood temperament at 18–27 months of age using the Early Childhood Behavior Questionnaire [20], to cognition at 2 years of age using the Mullen Scales of Early Learning [21], and to communication, motor, personal and social skills at 3 years of age using the Ages and Stages Questionnaire, Third Edition [22]. In this study, we examined whether the gut microbiota was associated with infant neurodevelopment in 71 full-term healthy infants aged 18 months. We used 16S rRNA gene sequencing for the analysis of the infants’ gut microbiota. The global neurodevelopment was assessed with the Bayley Scales of Infant and Toddler Development^®^, Third Edition (Bayley^®^-III) [23], the most common assessment tool used for global neurodevelopment before 2 years of age, that explores mental, motor and socioemotional development. We hypothesized that underlying community types or enterotypes defined by Dirichlet multinomial mixture modelling and taxa assessed by differential abundance analysis would vary between infants according to neurodevelopmental characteristics.

## 2. Materials and Methods

### 2.1. Subjects, Experimental Design and Ethical Guidelines

In the present study, 71 full-term healthy infants aged 18 months, who did not present any intestinal disorders and had not taken antibiotics, were chosen from the panel of infants that belonged to PREOBE observational study cohort [24]. In this period of life, the transition from weaning to solid food consumption occurs. General characteristics of the study population are shown in Appendix A. In this project, pregnant women were recruited between 2007 and 2012 at the San Cecilio and Mother-Infant University Hospitals in Granada, Spain. The study exclusion criteria for mothers were: simultaneous participation in any other research study, any kind of drug treatment, diagnosed diseases (e.g., hypertension or preeclampsia, intrauterine growth retardation, maternal infection, hypo/hyper- thyroidism, hepatic or renal disease), and vegan diet. Fresh stools were collected at 18 months after delivery and were immediately stored at −80 °C until processing. The study included anthropometric measurements, health questionnaires and medical assessments of the child. This project followed the ethical standards recognized by the Declaration of Helsinki (reviewed in Hong-Kong 1989 and in Edinburgh 2000) and the EEC Good Clinical Practice recommendations (document 111/3976/88 1990), and current Spanish legislation regulating clinical research in humans (Royal Decree 561/1993). The study was explained to the participants before starting, and the parents signed an informed consent form for themselves and provided written consent on behalf of their children.

### 2.2. DNA Extraction from Stool Samples

Genomic DNA was extracted from faecal bacteria of 18-month-old (*n* = 71) infants as previously described [25]. Briefly, faecal samples were resuspended in 1 mL of TN150 buffer (10 mM Tris-HCl pH 8.0 and 150 mM NaCl). Zirconium glass beads (0.3 g) and 150 mL of buffered phenol were added, and bacteria were disrupted with a mini bead beater set to 5000 rpm at 4 °C for 15 s (Biospec Products, Bartlesville, OK, USA). After centrifugation, genomic DNA was purified from the supernatant using phenol-chloroform extraction. Quality was checked by agarose gel electrophoresis and quantified with Quant-iT PicoGreen dsDNA assay kit (Invitrogen, Darmstadt, Germany). 

### 2.3. 16S rRNA Gene Sequencing and Data Processing

Genomic DNA from faecal bacteria was used as a template for 16S rRNA gene amplification using 27F and 338R universal primers and two consecutive PCR reactions to integrate Illumina multiplexing sequences as previously described [26]. The library was prepared by pooling equimolar ratios of amplicons and was sequenced using an Illumina MiSeq platform (Genetic Service, University of Granada, Granada, Spain). Reads were demultiplexed and sorted, and paired ends were matched to give 240 nt reads. The dataset was filtered and OTUs were defined at 99% similarity with MOTHUR programs unique.seqs and pre.cluster [27]. Taxonomic classifications of OTUs were assigned using the naïve Bayesian algorithm CLASSIFIER of Ribosomal Database Project [28]. OTUs were considered unassigned when the confidence value score was lower than 0.8, and were annotated using upper taxonomic ranks.

### 2.4. Assessment of Infant Cognitive Development

Infant cognitive development was assessed at 18 months of age using the Bayley Scales of Infant and Toddler Development^®^, Third Edition [23], by trained psychologists in the presence of the mother of the child. These scales measure the level of motor, language and cognitive or mental development. 

### 2.5. Statistical and Data Analysis

Statistical analyses were carried out using R statistical package [29]. a-diversity was measured at the OTU level using *phyloseq* package in R [30]. b-diversity for compositional data was calculated as Unifrac distance with *GUnifrac* package [31]. Quantifications of variances were calculated using PERMANOVA with the *adonis* function in the R package Vegan [32]. Dirichlet multinomial mixture (DMM) clustering is an unsupervised clustering method that uses Laplace approximation to identify groups of community assemblies or enterotypes at genus level [33]. The number of Dirichlet components was initially set to k = 4. Taxa unclassified at the genus level or present in less than 20% of the samples were excluded for DMM clustering. Fisher’s exact test was used to determine if infant FM groups were associated with enterotypes. Significant differential phylotype abundance at several different taxonomy levels was constructed from non-normalized raw count tables with the *DESeq2* package using a two-sided Wald test with adjustment for multiple comparisons by the Benjamini–Hochberg method [34]. For all determinations, the significance cut-off was set at *p* ≤ 0.05 or false discovery rate (FDR) ≤ 0.05 when multiple test correction was applied.

## 3. Results

### 3.1. Study Population Characteristics and Bayley^®^-III Scores

This study included 71 mother–infant pairs (45 boys and 26 girls) that were recruited in the PREOBE study cohort (Table 1). Bayley^®^-III scales of infant development were recorded in these full-term healthy infants at the age of 18 months (Appendix A). Children included in this study met the threshold for typical neurodevelopment and were dichotomized into two groups, above and below the median (50th percentile), according to their scores in three individual Bayley^®^-III domains: composite cognition, language (receptive language, expressive language and composite language) and motor (gross motor and fine motor). The medians (ranges) for Bayley^®^-III scores were as follows: composite cognition (125 (90–140)), receptive language (12 (8–15)), expressive language (11 (6–15)), composite language (106 (83–124)), gross motor (12 (7–12)) and fine motor (13 (8–19)).

### 3.2. Taxonomic Profiling of Infants’ Gut Microbiota

We collected faecal samples from healthy infants of 18 months of age to characterize the gut microbial composition by amplicon sequencing of the 16S rRNA hypervariable V1-V2 gene region. After quality filtering, 3,354,210 read sequences (mean per sample reads = 47,242; SD = 29,220) rendered a gut microbial profile consisting of 679 species-level bacterial operational taxonomic units (OTUs) that narrowed to 87 distinct genera belonging to 38 families after high confidence phylogenetic annotation. The phylogenetic composition and categorical breakdown of identified OTUs in our samples are presented in the supporting material (Appendix A). Taxonomic classification of OTUs was performed against RDP taxonomic database, resulting in a community membership dominated by taxa within Firmicutes (516), followed by Bacteroidetes (104), Proteobacteria (32), Actinobacteria (23) and Fusobacteria (4). Analysis of core microbes of all infants revealed twenty-one highly abundant OTUs (>1% of all sequence reads), but only six of them were detected in all samples. These prevalent and highly abundant OTUs belonged to the *Lachnospiracea*_*incertae*_*sedis*, *Streptococcus*, *Fusicatenibacter*, *Anaerostipes* and *Faecalibacterium* genera. At genus level, nineteen genera were highly abundant (>1% of all sequence reads) and accounted for 89.6% of total reads, with dominance of *Bacteroides*, *Lachnospiracea*_*incertae*_*sedis*, *unclass*_*Lachnospiraceae*, *Fusicatenibacter* and *Streptococcus* (each providing more than 5% of all sequence reads) (Figure 1A).

### 3.3. Gut Microbial Structure Is Different Regarding Fine Motor Skills

We initially examined whether measures of gut microbial community structure and diversity differed between infants categorized as above and below the median in the six individual Bayley^®^-III scores. Microbial a-diversity (intra-sample diversity), assessed by the number of taxa (richness), a phylogenetic diversity measurement (Faith’s phylogenetic diversity) and a non-phylogenetic measurement of bacterial abundance and evenness (Shannon’s diversity indexes), was not significantly different between above- and below-median groups for each Bayley^®^-III scale (Appendix A). Additionally, principal coordinate analyses based on UniFrac distance metrics was used to compare b-diversity (inter-sample diversity) between above- and below-median groups for each Bayley^®^-III scale. UniFrac measures the distance between microbial communities based on their abundance (weighted) and occurrence (unweighted) coupled to their phylogenetic relatedness. Considering all OTUs, the only Bayley^®^-III score with a strong association with gut microbial community structure and composition was FM skills, explaining 4% of microbiota variation (Figure 1B and Appendix A). Infants below-median clustered away from infants above-median using weighted (*p* = 0.021) but not unweighted (*p* = 0.552) UniFrac distance metrics (Appendix A). We also tested whether the gut microbiota b-diversity was associated with anthropometric, maternal and nutritional variables (Appendix A). The distance-based PERMANOVA test showed a significant explaining effect of maternal pregestational BMI (categorized as normoweight, overweight or obese) and type of breastmilk feeding up to the third month (formula, mixed or exclusive breastfeeding) on b-diversity using the weighted UniFrac distance matrix (Figure 1B,C). Interestingly, breastmilk feeding up to the third month (d) showed no effect on microbial community dissimilarities, suggesting that consumption of any breast milk impacted gut microbial assemblages in our cohort rather than the frequency of breastmilk feeding [35]. Together, FM, maternal pregestational BMI and type of breastmilk feeding up to the third month explained 13.7% of the variation in our dataset. No significant associations with maternal age, smoking during pregnancy (Yes/No), alcohol drinking during pregnancy (Yes/No), neonate weight, sex, type of delivery (C-section, vaginal) or gestational diabetes (Yes/No) were observed. These significant associations were considered in downstream statistical analyses to avoid covariate effects.

### 3.4. Enterotypes Are Associated with Fine Motor Activity in Infants

To investigate the potential association of FM scores with the prevalence of gut microbial community profiles, we identified community types in infant’s samples by using Dirichlet multinomial mixtures (DMM) modelling that assigned them into clusters based on the contribution of taxa at the genus level. Model fitting rendered an optimum number of two DMM community types, here referred to as enterotypes (Figure 2A). The two gut microbial enterotypes had weights *p* = 0.78 and 0.22 and variability Θ = 15.45 and 51.36. Thus, the highly abundant enterotype (78% of samples) comprised samples with more variable community profiles, while the less prevalent enterotype (22% of samples) comprised samples with more similar community assemblages (Figure 2B). We observed that enterotypes were classified by thirty genera that accounted for 85.4% of total reads in our dataset, with mean total reads ranging from 0.07% (*Flavonifractor*) to 21.3% (*Bacteroides*) (Figure 2C). Highly abundant genera (>1% of total reads) such as *unclass*_*Clostridiales*, *unclass*_*Ruminococcaceae*, *Ruminococcus*, *unclass*_*Peptostreptococcaceae*, *unclass*_*Enterobacteriaceae*, *Enterococcus*, *Faecalibacterium*, *Ruminococcus2* and *Roseburia* did not drive enterotype classification. A core set of nine genera were the most important drivers in model prediction (Figure 2D). The first enterotype contained samples with mixed microbial composition whose most important drivers were a group of genera within *Firmicutes* such as *Lachnospiracea*_*incertae*_*sedis*, *unclass*_*Lachnospiraceae*, *Streptococcus* and *Blautia*, and specific contribution by *Fusicatenibacter* and *Anaerostipes* (Figure 2D). The second enterotype was mostly driven by *Bacteroides* with specific contribution of *Clostridium* XIVa and *Parabacteroides* to enterotype assignment. On the basis of their respective genus-level dominance profiles, we referred to enterotypes as *Firmicutes* dominant type (Firm) and *Bacteroides* dominant type (Bact). Signature genera of each enterotype classification accounted for 81.7% (Bact; SD = 7.2) and 68.8% (Firm; SD = 18.5) of the mean total abundances in their corresponding samples (Figure 2E). These results show that signature genera accounted for most reads in the Bact enterotype, with *Bacteroides* as the main contributor to overall enterotype assemblage, whereas signature *Lachnospiracea*_*incertae*_*sedis* and *unclass*_*Lachnospiraceae* were the most abundant genera in the more variable Firm enterotype. Finally, to test for associations between enterotypes and FM, we performed a Fisher’s exact two-tailed test. Belonging to an enterotype was significantly associated with an infant’s FM group, where above-median samples belonged to the *Firm* enterotype and below-median group mostly belonged to the *Bact* enterotype (*p* = 0.04, odds ratio = 0.27).

### 3.5. Thirteen Genera including Bifidobacterium and Lactobacillus Differentiated Fine Motricity Groups

To further investigate the association of fine motricity with microbiota composition, we used the statistical software DESeq2 and adjusted the model for covariates to identify taxa at the genus level with differential abundances between below- and above-median FM groups. We identified 11 genera overabundant in the above-median group and 2 genera in the below-median group (Figure 3). These overabundant genera were phylogenetically different. Discriminating genera belonged to *Actinobacteria*, *Firmicutes* and *Fusobacteria* in above-median FM infants, and to *Bacteroidetes* and *Firmicutes* in below-median FM infants. The gut microbiota of above-median FM infants showed a high prevalence of *Bifidobacterium*, *Collinsella*, *Coprococcus*, *Enterococcus*, *Fusobacterium*, *Holdemanella*, *Lactobacillus*, *Propionibacterium*, *Roseburia*, *Veillonella* and an unassigned genus within *Veillonellaceae*. In contrast, the gut microbiota of below-median FM infants was enriched in *Parabacteroides* and *Turicibacter*. Being probiotic *Bifidobacterium* and *Lactobacillus* enriched in above-median FM infants, we conducted a more detailed analysis on the association of microbial species belonging to *Bifidobacterium* (7 OTUs) and *Lactobacillus* (11 OTUs) genera with FM. The analysis revealed that only *Bifidobacterium* OTU-242 tended to be enriched in infants with above-median FM (Log_2_FC 2.51, FDR < 0.05). Both BLASTN and EzTaxon nucleotide sequence comparisons showed that OUT-242 had 100% sequence identity with the 16S rRNA of *B. bifidum* ATCC 29521.

## 4. Discussion

There is increasing agreement among human and animal studies that the gut microbiota impacts brain development, neurological outcomes and disorders, resulting in long-term changes in behaviour that we are only beginning to appreciate. Seminal studies in germ-free mice emphasized that early colonization with a complete specific-pathogen-free microbiota ameliorated brain development and behavioural abnormalities [11,14,36]. Cross-sectional clinical studies associated altered microbial composition with the pathophysiology of Alzheimer’s disease, autism spectrum disorder, multiple sclerosis, Parkinson’s disease, and stroke [37]. Critically important is understanding the behavioural effects of gut microbial colonization during early life, the most rapid and dynamic phase of postnatal brain development [38]. Moreover, bacterial composition begins to converge toward an adult-like microbiota by the end of the first year of life and fully resembles the adult microbiota by 2.5 years of age. A mid-time point where microbiota and neurodevelopment are still converging would probably be better to observe differences between infants. Data on the direct connection between gut microbiota and infant neurodevelopment are lacking. In the current study, we studied the association of gut microbiota structure assessed by a-diversity (within subject) and b-diversity (between subjects) metrics with neurodevelopment at the cognitive, language and motor levels in a cohort of healthy developing 18-month-old infants. We further sought to identify specific signature taxa within microbial profiles or enterotypes. This is the first study to describe associations between gut microbiota enterotypes with fine motor skills in infants and to identify significantly different signature taxa. Importantly, associations between enterotypes and fine motor scores were present when controlling for breastfeeding and maternal pregestational BMI as covariates.

Almost certainly considered by the field as the gold standard, the Bayley^®^-III is currently the most widespread individually administered test for behavioural assessment of children and young children aged 0 to 42 months, used in both clinical and research settings [23]. Conducted by an expert examiner, the structure of the Bayley^®^-III, which is divided into distinct composite scores (Cognitive, Language and Motor), allows a more precise assessment of specific developmental domains, identifying children with developmental delays and to provide information for intervention planning. In our study, children with above or below median values for each Bayley^®^-III subscale did not differ substantially in a-diversity, as indicated by community richness, evenness, phylogenetic similarity and species abundances. Our results contrast what was found by Carlson et al. [21], where higher a-diversity was associated with lower scores on the overall composite score, visual reception scale, and expressive language scale at 2 years of age using the Mullen Scales of Early Learning. However, our findings agree with the study of Sordillo et al. in 3 year old children [22]. The authors used the Ages and Stages Questionnaire, Third Edition, and reported no associations between gut microbiota a-diversity and developmental outcomes. Consistent associations of microbiota b-diversity with infant gut microbiota were observed for FM subscale in our study. FM abilities refer to the control of small muscles that is highly related to cognitive functions that process visual information [39]. Despite this connection, examination of Bayley^®^-III scales showed no significant associations of cognitive and language subscales with gut microbiota in our cohort. The significant association between microbiota b-diversity and FM groups was observed using weighted UniFrac distances and were not paralleled with unweighted UniFrac distances. This result indicates that differences in OTU abundances rather than in the presence and absence of OTUs are associated with the overall community structure in FM groups. Differences in b-diversity in healthy infants have only been reported by Christian et al. [20], who observed significant associations between the gut microbiota and temperament in boys. In our study, child sex had no impact on gut microbiota structure.

In order to reveal plausible underlying microbial assemblages in FM groups, we enterotyped our study cohort using Dirichlet multinomial mixtures on genus-level OTU profiles. Based on the genus-level proportional abundances, two distinct enterotypes were identified and termed Bact and Firm due to the dominant contribution of *Bacteroides* and *Firmicutes* in each enterotype. Our enterotype analysis aligned with previous reports of microbiota variation centred around *Bifidobacterium*, *Bacteroides*, *Prevotella* and *Ruminococcaceae* [40,41]. The first 2 years of life are characterized by a succession of bacterial taxa in which a facultative aerobic Enterobacteriaceae-rich community evolves into a strict anaerobe-rich community containing *Bacteroides* that is progressively enriched in a diverse mixture of *Ruminococcaceae*, *Faecalibacterium*, and *Lachnospira* within *Firmicutes* [42]. Of note, a statistically significant association between the prevalence of Bact and Firm enterotypes with fine motricity skills was observed. This is the first published prospective infant study linking enterotypes and infant neurodevelopment, showing poorer performances of infants on FM with a *Bacteroides*-dominant community. Our results confirm the observations by Sordillo et al. on the association of *Bacteroides*-dominated co-abundance grouping with poorer FM scores [22]. In addition, we identified taxa whose abundance was significantly different between FM groups. The motricity of below-median FM group was associated with increased abundances of *Turicibacter* and *Parabacteroides*. This is the first report of an association between these taxa and host behaviour in healthy infants. Finegold et al. established a list of significant genera among severely autistic vs. non-sibling control samples, where *Turicibacter* and *Parabacteroides* were significantly increased in autistic children [17]. Motor coordination impairment is a common condition in children with neurodevelopmental disorders such as autism [43]. In our cohort, better fine motor skills were associated with increased abundances of eleven genera, including *Coprococcus* and an unassigned genus within *Veillonellaceae*. Kang et al. [44] reported that *Coprococcus* and *unclass*_ *Veillonellaceae* were less abundant in autistic individuals compared with neurotypical controls between the ages of 3 and 16 years. An interesting finding in the above-median FM group was an over-abundance of *Lactobacillus* and *Bifidobacterium*, the most important and widely used probiotic genera. Several studies explored the role of these probiotics in the gut–brain axis. Bravo et al. observed that the ingestion of *L. rhamnosus* (JB-1) regulated emotional behaviour and central GABA receptor expression in mice via the vagus nerve [45]. In another study, Davari et al. reported that a supplementation of *L. acidophilus*, *B. lactis* and *L. fermentum* improved diabetes-induced impairment of synaptic activity and cognitive function in rats [46]. Still, more studies on rodent–human translation are required. A well-known study from Messaoudi et al. revealed a reduced self-reported anxiety in humans treated with *L. helveticus* R0052 and *B. longum* R0175 [47]. Similar effects have been observed in other investigations of mood. For instance, in a recent randomised controlled trial [48], healthy male and female participants consumed either a placebo product or a mixture of several probiotics (*B. bifidum* W23, *B. lactis* W52, *L. acidophilus* W37, *L. brevis* W63, *L. casei* W56, *L. salivarius* W24, and *L. lactis* W19 and W58). Relative to placebo, probiotic-treated participants exhibited substantially reduced reactivity to a sad mood (assessed by the Leiden Index of Depression Sensitivity Scale). Therefore, administration of probiotics independently or in combination may positively affect motor skills in early childhood [49].

## 5. Conclusions

We observed a significant association between gut microbiota and fine motricity skills in 18-month-old full-term healthy infants, robust to adjustment for infant, mother, nutritional and perinatal variables. Our study adds to the mounting evidence connecting the gut microbiota with the gut–brain axis, where the initial stages of gut colonization and assemblage may be linked with neurodevelopmental outcomes with potential long-term associations. These results suggest targeting for negatively associated species such as *Turicibacter* and *Parabacteroides* in appropriate mice models, and invite further interventional studies using *Lactobacillus* and *Bifidobacterium* strains to influence motricity outcomes in infants.

## Figures and Tables

**Figure 1 nutrients-13-01673-f001:**
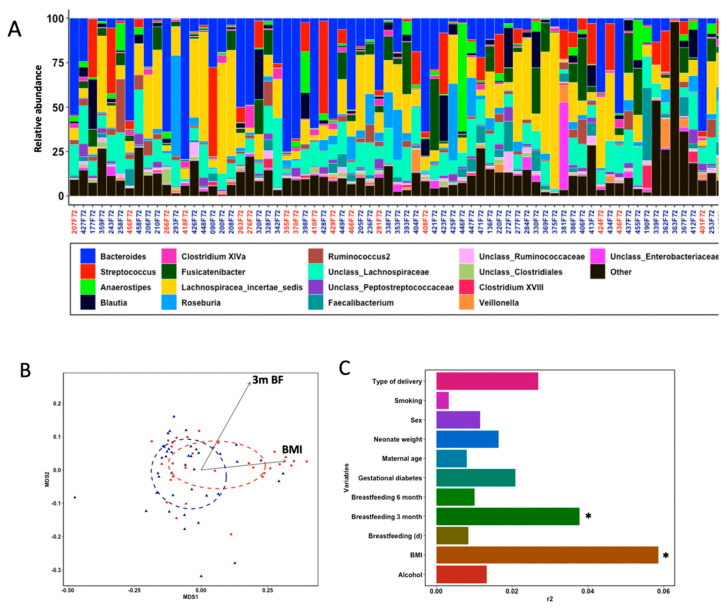
Gut microbial composition and structure in 71 full-term healthy infants at 18 months of age. (**A**) Phylogeny of infants’ gut microbiota at the genus level of 17 core genera detected in ≥98% of study subjects. Sample names are coloured according to enterotype: Firm in blue, Bact in red. Gut microbial composition and structure in 71 full-term healthy infants at 18 months of age. (**B**) Scatterplot from principal coordinate analysis using weighted Unifrac metrics in above-median (blue) and below-median (red) fine motor groups. Additional significant explanatory variables are represented by black arrows originating from the coordinate system origin. (**C**) Horizontal bars show the influence of anthropometric, maternal and nutritional variables (r^2^) on gut microbiota composition. * *p* < 0.05 for PERMANOVA test with 999 permutations.

**Figure 2 nutrients-13-01673-f002:**
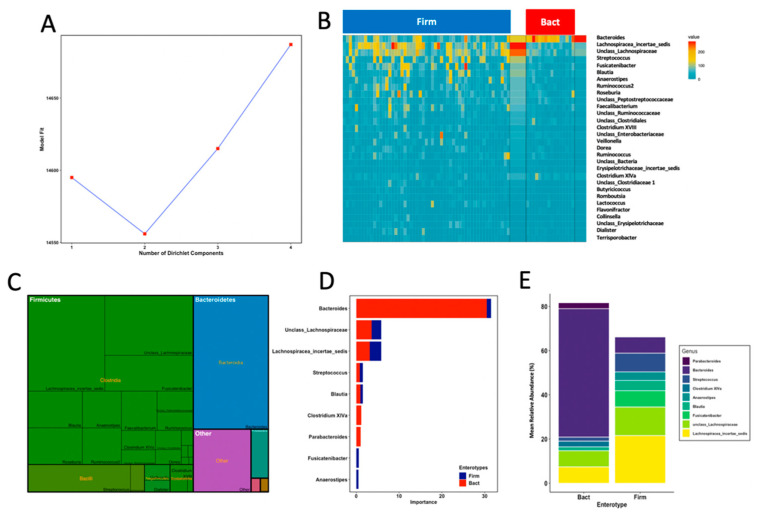
Probabilistic modelling with Dirichlet multinomial mixtures of faecal samples revealed 2 enterotypes in 71 full-term healthy infants at 18 months of age. (**A**) The model was fit to k = 4 and rendered the lowest model fit value of Laplace approximation for four components which was taken as evidence for the presence of the two main community types. (**B**) Heatmap showing the relative abundance of the 30 most dominant genera per DMM enterotype. Abundance represents square root sequence reads. (**C**) Contribution of the 30 most dominant genera to overall community composition. (**D**) Contribution to model prediction of microbial core genera to each enterotype. (**E**) Contribution of core signature genera (mean relative abundances) to microbial profiles in each DMM enterotype.

**Figure 3 nutrients-13-01673-f003:**
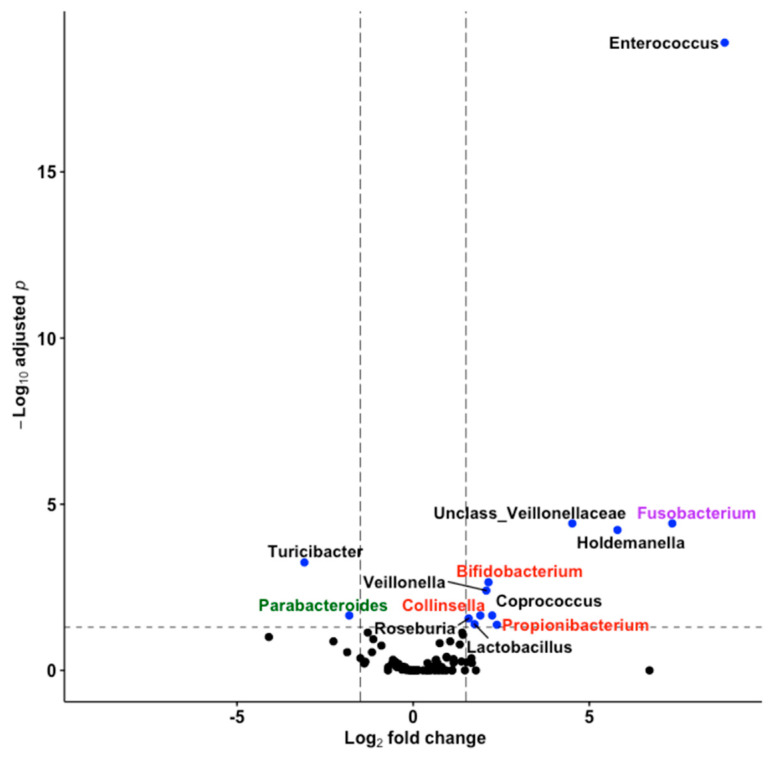
Differential abundance of gut microbiota between infants with above- and below-median fine motricity groups. Volcano plot showing shrunken log_2_ fold changes in mean abundance of genera versus −log_10_ of FDR. Blue colours depict significance and label names denote the taxonomy of genera at phylum level: *Firmicutes* (black), *Actinobacteria* (red), *Bacteroidetes* (green), *Fusobacteria* (purple). Dashed lines denote thresholds of significance (FDR < 0.05).

**Table 1 nutrients-13-01673-t001:** Characteristics of the population according to the dichotomization between Fine Motricity Scores from the Bayley^®^-III.

Characteristic	Below the Median	Above the Median	*p*
**Maternal characteristics**			
**Age (years)**	32.87 (5.37)	33.57 (2.97)	0.485
**Pre-pregnancy BMI**			0.034 *
Normal-Weight	12 (38.71%)	9 (22.5%)	
Overweight	9 (29.03%)	24 (60.0%)	
Obesity	10 (32.26%)	7 (17.5%)	
**Pre-pregnancy weight (kg)**	72.12 (14.3)	72.38 (15.17)	0.966
**Weight gain (kg)**	8.71 (4.23)	8.35 (7.62)	0.832
**Diabetes**			0.515
No	17 (54.84%)	25 (62.5%)	
Yes	14 (45.16%)	15 (37.5%)	
**Education**			0.154
Primary/Secondary	20 (64.52%)	18 (45.0%)	
University	11 (35.48%)	20 (50.0%)	
**Smoking during pregnancy**			0.744
No	28 (90.32%)	37 (92.5%)	
Yes	3 (9.68%)	3 (7.5%)	
**Drinking alcohol during pregnancy**			0.855
No	30 (96.77%)	39 (97.5%)	
Yes	1 (3.23%)	1 (2.5%)	
**Type of delivery**			0.644
Eutocic	20 (64.52%)	22 (55.0%)	
Dystocic	4 (12.90%)	5 (12.5%)	
Cesarean	7 (22.58%)	13 (32.5%)	
**Infant characteristics**			
Sex			
Male	22 (70.97%)	23 (57.5%)	0.243
Female	9 (29.03%)	17 (42.5%)	
**Birth weight (g)**	3253.87 (467.99)	3287.25 (578.51)	0.794
**Birth length (cm)**	50.68 (1.96)	49.83 (2.02)	0.098
**Birth head Circumference (cm)**	34.75 (1.29)	34.66 (1.47)	0.803
**Placenta (g)**	498.52 (108.69)	532.06 (127.56)	0.281
**Breastfeeding ^†^**			0.032 *
Exclusive	14 (45.16%)	29 (72.5%)	
Mixed	4 (12.90%)	5 (12.5%)	
Artificial	13 (41.94%)	6 (15.0%)	

Values listed are the total for the variable (percent of total value *n*) or means (SD). Differences between Bayley^®^-III Fine Motricity groups according to T-Student’s test. Chi-square test was applied to qualitative variables. * *p*-values ≤ 0.05. ^†^ Breastfeeding practice information was collected at 3 months of age of the child.

## Data Availability

The data presented in this study are available on request from the corresponding author.

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
