# Peer review of "Infant Gut Microbiota Associated with Fine Motor Skills"

_nutrients, 2021, doi:10.3390/nu13051673_

Round 1

Reviewer 1 Report

Comments: The manuscript is well written and suitable for publication, however I would recommend the authors to consider the below suggestions and comments.

Methods:

Use of the Bayley-III tool: Did the study team use the full tool, including the parental report for social-emotional and adaptive behaviour domains?

Chosen age for stool collection and neurodevelopment assessment: is “in this period of life, the transition from weaning to solid food consumption occurs” the reason and only reason for choosing the time point of 18 months?

Results:

Line 147: do the authors mean “all” children?

Lines 151-153: The Bayley-III tool has been developed to detect developmental delays. The scores from this population was within the normal range (which should be mentioned). It is therefore questionable to divide the population between below and above the median when all toddlers had normal developmental scores. Could the authors comment further?

Discussion:

There should be further discussion on the effect of population characteristics and differences between the two groups that may influence the gut microbiota, in particular the positive effect of exclusive breastfeeding and duration of breastfeeding. For example, in lines 352-354, the authors should comment on the potential effect of exclusive breastfeeding on the abundance of Lactobacillus and Bifidobacterium. The conclusion should also promote the role of breastfeeding and healthy maternal weight for the development of infants.

References: please add He et al. DOI: 10.1038/s41598-019-47953-4

Minor edits:

The naming convention for the neurodevelopment tool is Bayley Scales of Infant and Toddler Development®, Third Edition (Bayley®-III). Please change throughout the manuscript.

Line 80: please mention that the PREOBE is an observational study.

Line 81: reference to table 1 is not ideal as the study population in Table 1 is split between 2 groups and does not give a representation of the entire cohort. A reference to the original study article or add a supplementary table would be preferrable.

Table 1: The title, the headings and the footnote need reworking to make it easier for the reader to read and understand the table.

Line 108: please correct “16S”

Please remove line 144-145 as it is already stated in the method section.

Tables S2 and S3: please change the title to reflect what is presented (i.e. correlation/association…).

Section 3.3: “-diversity” is missing a letter at the start. Line 191: please remove the interrogation mark. Line 189: please change the word “unique” to “only” and line 190: “activity” to “skills”.

Lines 198-200: the sentence is confusing and should be rewritten.

Line 304: a word is missing after “divided”.

Line 308: please add the reference after “Carlson et al.”

Line 369: please shift the ]

Author Response

COMMENTS FOR THE AUTHOR:

Title: Infant Gut Microbiota Associated with Fine Motor Skills.

Manuscript ID: nutrients-1188578

Reviewer #1: Comments:

The manuscript is well written and suitable for publication, however I would recommend the authors to consider the below suggestions and comments.

We wish to acknowledge that the typeface input worked but we do not know why it was changed and came out distorted.

Methods:

- Use of the Bayley-III tool: Did the study team use the full tool, including the parental report for social-emotional and adaptive behaviour domains?

Yes, parents answered to nutritional, life style, clinical and socio-emotional questionnaire.

- Chosen age for stool collection and neurodevelopment assessment: is “in this period of life, the transition from weaning to solid food consumption occurs” the reason and only reason for choosing the time point of 18 months?

The PREOBE project collected stool samples at several time points to assess microbiota development since birth. Bacterial composition begins to converge toward an adult-like microbiota by the end of the first year of life and fully resembles the adult microbiota by 2.5 years of age. We chose a mid-time point where development was still converging and differences would be probably better observed between infants. We included this in the Discussion of the manuscript (lines 289-293).

Results:

- Line 147: do the authors mean “all” children?

For statistical test, yes. All children included in this study (n=71) met the threshold for typical neurodevelopment.

- Lines 151-153: The Bayley-III tool has been developed to detect developmental delays. The scores from this population was within the normal range (which should be mentioned). It is therefore questionable to divide the population between below and above the median when all toddlers had normal developmental scores. Could the authors comment further?

As stated, we studied full-term healthy infants with no neurodevelopmental disorder. Thus, skill scores lie within the normal range. We agree that this point should be underscored. We included the data in Table S2.

Despite being healthy, scores range widely and dichotomization is achievable to generate sensible study groups.

Discussion:

- There should be further discussion on the effect of population characteristics and differences between the two groups that may influence the gut microbiota, in particular the positive effect of exclusive breastfeeding and duration of breastfeeding. For example, in lines 352-354, the authors should comment on the potential effect of exclusive breastfeeding on the abundance of Lactobacillus and Bifidobacterium. The conclusion should also promote the role of breastfeeding and healthy maternal weight for the development of infants.

We agree that breastfeeding and healthy maternal weight is important for the development of infants. The discussion was centered on the association between gut microbiota and motor skills because effect of these covariates was removed to demonstrate statistical significance. Thus, associations with FM are robust. Including the effect of breastfeeding and BMI would eclipse our findings.

References:

- Please add He et al. DOI: 10.1038/s41598-019-47953-4

Thank you for your advice. We have added this reference to our manuscript.

Minor edits:

- The naming convention for the neurodevelopment tool is Bayley Scales of Infant and Toddler Development®, Third Edition (Bayley®-III). Please change throughout the manuscript.

Done

- Line 80: please mention that the PREOBE is an observational study.

Done

- Line 81: reference to table 1 is not ideal as the study population in Table 1 is split between 2 groups and does not give a representation of the entire cohort. A reference to the original study article or add a supplementary table would be preferrable.

We thank the Reviewer for the comments. We have added a supplementary table collecting the characteristics of the entire cohort (Table S1).

- Table 1: The title, the headings and the footnote need reworking to make it easier for the reader to read and understand the table.

Done

- Line 108: please correct “16S”

Done

- Please remove line 144-145 as it is already stated in the method section.

Done

- Tables S2 and S3: please change the title to reflect what is presented (i.e. correlation/association…).

Done. Former Tables S2 and S3 are currently Tables S4, S5 and S6.

- Section 3.3: “-diversity” is missing a letter at the start. Line 191: please remove the interrogation mark. Line 189: please change the word “unique” to “only” and line 190: “activity” to “skills”.

Done

- Lines 198-200: the sentence is confusing and should be rewritten.

There was a mistake

We also tested whether the gut microbiota b-diversity of FM groups was associated to anthropometric, maternal and nutritional variables (Table S4).

Has been replaced by

We also tested whether the gut microbiota b-diversity was associated to anthropometric, maternal and nutritional variables (Table S7).

- Line 304: a word is missing after “divided”.

Done

- Line 308: please add the reference after “Carlson et al.”

Done

- Line 369: please shift the ]

Done, thank you for your comments.

Reviewer 2 Report

The authors conducted an observational study to investigate the association between gut microbiota composition and neurodevelopment of infant aged 18 months. They found that the weighted UniFrac distance of gut microbiota showed significant difference in fine motor activity evaluated by BSID-III. The topic of this study is attractive for readers, however, I can seldom access one of the most important data, BSID-III scores in this manuscript. I feel this ambiguity raised some concerns. Some typos also confuse me to understand what they insisted.

Major comments

I understand the authors have divided 71 mother-infant pairs into two groups based on the median of BSID-III scores, but I cannot assess this dichotomization was suitable for the purpose of this study, because no information of the distribution of each score (as subject backgrouds) was indicated. I recommend to show the data of BSID-III scores not only median and ranges written at line 152-153.

If the BSID-III scores are continuous variable even if these are non-parametric, correlation analysis would be better than inter group comparison by DESeq2 shown in Figure 3.

The authors suggested enterotype was associated with FM activity shown in Figure 2 and related sentences. Although this suggestion seems correct at first glance, but I would say this is over-exaggerating. Because the following data in Figure 3 shows bacteria different from the main contributor of enterotype. Please consider to remove all result and discussion related to enterotype (mainly Figure 2).

Please describe why infant aged 18 months was selected for this study. No obvious reason was found in the introduction.

Line 166, it was described “six OTUs were detected in all samples”, but the following sentence explains only five bacteria taxa. Similarly, at line 169, nineteen genera were highly abundant, but Figure 1 shows only 17 core genera. Such kind of inconsistency should disturb readers’ understanding.

Table S2 shows what? p-values? It would be better to show mean+ SD of alpha diversities in two groups.

In my understanding, both of Table S3 and S4 show the result by PERMANOVA, but why table S3 only shows p-value ?

I think the best knowledge of this study was to find the association between gut microbiota and FM activity. Why Figure 1C shows mainly other variables, but not FM activity (e.g. table S3)?

Line 202, a total of FM, maternal pregestational BMI and type of breastmilk feeding up to the third month explained 13.7% of variation in our dataset. I could not find the FM data, so please indicate the evidence of the 13.7 %.

Line 359, I don’t think the rodent-human translation has been surprisingly robust at least gut microbiota area. For example, genus Bifidobacterium is dominant in human but uncommon in experimental animals.

Minor comments

Line 191, ? probably should be removed

Line193,197. Beta-diversity ?

It is better to large the symbols in Figure 1B.

Line 271, FDR<0.06 ?

reference 495 is probably 45

Author Response

COMMENTS FOR THE AUTHOR:

Title: Infant Gut Microbiota Associated with Fine Motor Skills.

Manuscript ID: nutrients-1188578

Reviewer #2: Comments:

The authors conducted an observational study to investigate the association between gut microbiota composition and neurodevelopment of infant aged 18 months. They found that the weighted UniFrac distance of gut microbiota showed significant difference in fine motor activity evaluated by BSID-III. The topic of this study is attractive for readers, however, I can seldom access one of the most important data, BSID-III scores in this manuscript. I feel this ambiguity raised some concerns. Some typos also confuse me to understand what they insisted.

Major comments

- I understand the authors have divided 71 mother-infant pairs into two groups based on the median of BSID-III scores, but I cannot assess this dichotomization was suitable for the purpose of this study, because no information of the distribution of each score (as subject backgrouds) was indicated. I recommend to show the data of BSID-III scores not only median and ranges written at line 152-153.

We agree that this point should be underscored. We have included the data in Table S2.

- If the BSID-III scores are continuous variable even if these are non-parametric, correlation analysis would be better than inter group comparison by DESeq2 shown in Figure 3.

- The authors suggested enterotype was associated with FM activity shown in Figure 2 and related sentences. Although this suggestion seems correct at first glance, but I would say this is over-exaggerating. Because the following data in Figure 3 shows bacteria different from the main contributor of enterotype. Please consider to remove all result and discussion related to enterotype (mainly Figure 2).

For both comments, our answer is the following:

Our view is to consider the microbiota as an ecosystem rather than the mere listing of species. Weighted UniFrac indicated an underlying association between microbiota abundance and motor skills. After enterotype classification, significant statistical association was observed between enterotypes and motor skills as indicated in the manuscript (line 330-331). Structures point to microbial co-abundances or exclusions, a step-beyond, where communities are built by a network of within-species interactions, an ecosystem defined by an enterotype, allowing further comparisons with other cohorts and

stages of human development. Once you have a holistic view of the community defined, DESEq seeks for sub-structural associations between genera and FM groups, that is, taxa that do not necessarily belong to enterotype driver genera. We sum ecology to microbiology.

- Please describe why infant aged 18 months was selected for this study. No obvious reason was found in the introduction.

The PREOBE project collected stool samples at several time points to assess microbiota development since birth. Bacterial composition begins to converge toward an adult-like microbiota by the end of the first year of life and fully resembles the adult microbiota by 2.5 years of age. We chose a mid-time point where development was still converging and differences would be probably better observed between infants. We included this in the Discussion of the manuscript (lines 289-293).

- Line 166, it was described “six OTUs were detected in all samples”, but the following sentence explains only five bacteria taxa. Similarly, at line 169, nineteen genera were highly abundant, but Figure 1 shows only 17 core genera. Such kind of inconsistency should disturb readers’ understanding.

From the six OTUs two belong to the same genera, that makes five genera.

Rephrasing:

Analysis of core microbes of all infants revealed twenty-one highly abundant OTUs (>1% of all sequence reads) but only six of them were detected in all samples. These prevalent and highly abundant OTUs belonged to Lachnospiracea_incertae_sedis, Streptococcus, Fusicatenibacter, Anaerostipes and Faecalibacterium genera.

Line 169 rephrasing:

At genus level, nineteen genera were highly abundant (>1% of all sequence reads) and accounted for 89,6% of total reads, with dominance of Bacteroides, Lachnospiracea_incertae_sedis, unclass_Lachnospiraceae, Fusicatenibacter and Streptococcus (providing each more than 5% of all sequence reads) (Figure 1A).

- Table S2 shows what? p-values? It would be better to show mean+ SD of alpha diversities in two groups.

Done. Former Table S2 is currently Table S4

- In my understanding, both of Table S3 and S4 show the result by PERMANOVA, but why table S3 only shows p-value?

Thank you for your contribution. We have corrected all these problems. We have added the required information to the tables. Former Tables S3 and S4 are currently Tables S5, S6 and S7.

- I think the best knowledge of this study was to find the association between gut microbiota and FM activity. Why Figure 1C shows mainly other variables, but not FM activity (e.g. table S3)?

We showed the differences in microbiota structure according to FM groups in figure 1B and added the explained variation in the main text and Table S5 (former Table S3).

Figure 1C shows the rest of study variables.

Rephrasing:

Considering all OTUs, the only Bayley®-III score with a strong association with gut microbial community structure and composition was FM skills, explaining 4% of microbiota variation (Figure 1B and Table S5).

- Line 202, a total of FM, maternal pregestational BMI and type of breastmilk feeding up to the third month explained 13.7% of variation in our dataset. I could not find the FM data, so please indicate the evidence of the 13.7 %.

FM results for the weighted UniFrac are shown in Table S5. Figure 1C and Table S7 show the rest of study variables. Summing FM, maternal pregestational BMI and type of breastmilk feeding up to the third month is 13.7 % of the variation explained.

- Line 359, I don’t think the rodent-human translation has been surprisingly robust at least gut microbiota area. For example, genus Bifidobacterium is dominant in human but uncommon in experimental animals.

There is a wealth of knowledge on associations between probiotics and beneficial health effects in humans and rodents. The translation of phenotypes by fecal transfer between humans and mice is robust. Experimental problems arise when effect of probiotics is to be demonstrated because rodent gut discard most of them. Still, Bifidobacterium longum is consistently identified in human to rodent experiments. We agree that the probiotics concept solely with Lactobacillus and Bifidobacterium requires ecosystem rewriting.

Rephrasing:

Still, more studies on the rodent-human translation are required.

Minor comments

- Line 191, ? probably should be removed

Done

- Line193,197. Beta-diversity ?

Done. We wish to acknowledge that the typeface input worked but we do not know why it was changed and came out distorted.

- It is better to large the symbols in Figure 1B.

Done

- Line 271, FDR<0.06 ?

It was a mistake. We have corrected it for FDR<0.05.

- Reference 495 is probably 45

Done, thank you for your comments.

Round 2

Reviewer 2 Report

The authors have addressed all of my comments. From my point of view, the paper is now suitable for publication.